# Influence of Florfenicol Treatments on Marine-Sediment Microbiomes: A Metagenomic Study of Bacterial Communities in Proximity to Salmon Aquaculture in Southern Chile

**DOI:** 10.3390/antibiotics14101016

**Published:** 2025-10-13

**Authors:** Sergio Lynch, Pamela Thomson, Rodrigo Santibañez, Ruben Avendaño-Herrera

**Affiliations:** 1Facultad de Recursos Naturales y Medicina Veterinaria, Medicina Veterinaria, Universidad Santo Tomás, Puerto Montt 5480000, Chile; 2Interdisciplinary Center for Aquaculture Research (INCAR), Viña del Mar 2531015, Chile; pamela.thomson@unab.cl; 3School of Veterinary Medicine and One Health Institute, Faculty of Life Sciences, Andrés Bello University, Santiago 8370134, Chile; 4Facultad de Ingeniería, Departamento de Ingeniería Química y Bioprocesos, Pontificia Universidad Católica de Chile, Santiago 7820436, Chile; 5Laboratorio de Patología de Organismos Acuáticos y Biotecnología Acuícola, Facultad de Ciencias de la Vida, Universidad Andrés Bello, Viña del Mar 2531015, Chile; 6Centro de Investigación Marina Quintay (CIMARQ), Universidad Andrés Bello, Quintay 2340000, Chile

**Keywords:** florfenicol impact, marine benthic ecosystems, microbial diversity, antimicrobial

## Abstract

**Background/Objectives**: Metagenomic analyses are an important tool for understanding ecological effects, particularly in sites exposed to antimicrobial treatments. Marine sediments host diverse microbial communities and may serve as reservoirs for microbial resistance. Although it is known that antimicrobials can alter microbial composition, specific impacts on sediments surrounding salmon farms remain poorly understood. This study analyzed bacterial community structure in marine sediments subjected to florfenicol treatment from salmon farms in the Los Lagos Region of southern Chile. **Methods**: Sediment samples were collected and examined through DNA extraction and PCR amplification of the 16S rRNA gene (V3-V4 region). Sequences were analyzed using a bioinformatics pipeline, and amplicon sequence variants (ASVs) were taxonomically classified with a Naïve Bayesian classifier. The resulting ASV abundance were then used to predict metabolic functions and pathways via PICRUSt2, referencing the MetaCyc database. **Results**: Significant differences in bacterial phyla were observed between the control farm and two farms treated with florfenicol (17 mg kg^−1^ body weight per day) for 33 and 20 days, respectively. Farm 1 showed notable differences in phyla such as Bacteroidota, Bdellovibrionota, Crenarchaeota, Deferrisomatota, Desulfobacterota, Fibrobacterota, Firmicutes, and Fusobacteriota, while Farm 2 exhibited differences in the phyla Bdellovibrionota, Calditrichota, Crenarchaeota, Deferrisomatota, Desulfobacterota, Fusobacteriota, Nanoarchaeota, and Nitrospirota. Shannon Index analysis revealed a reduction in alpha diversity in the treated farms. Comparative analysis between the control and the treated farms showed pronounced shifts in the relative abundance of several bacterial phyla, including statistically significant differences in Chloroflexi and Firmicutes. Predicted functional pathways revealed a notable enrichment of L-methionine biosynthesis III in Farm 2, suggesting a shift in sulfur metabolism potentially driven by antimicrobial treatment. Additionally, increased activity in fatty acid oxidation pathways indicates a higher microbial potential for lipid degradation at this site. **Conclusions**: These findings highlight the considerable influence of florfenicol on sediment microbial communities and reinforce the need for sustainable management strategies to minimize ecological disruption and the spread of antimicrobial resistance.

## 1. Introduction

Marine sediments are a fundamental component of the oceanic ecosystem, serving both as a repository for organic matter and as a habitat for diverse microbial communities, the structure of which is shaped by multiple factors that vary across space and time [1]. These sediments play a crucial role in biogeochemical cycles, influencing nutrient dynamics, carbon sequestration, and the detoxification of pollutants [2]. The microbial communities within these sediments are pivotal in maintaining ecological balance, contributing to nutrient recycling and organic matter decomposition [3]. Understanding the metabolic processes and interactions of sediment microorganisms that shape microbial community dynamics using advanced high-throughput techniques, such as metagenomics, has proven powerful in linking microbial community structure to biogeochemical transformations in ecosystems [4].

Fish aquaculture, particularly salmon farming, has become a key contributor to global food production, offering a sustainable source of protein to meet the demands of a growing population [5]. However, the rapid expansion of aquaculture has raised environmental concerns, especially regarding the use of antimicrobials to manage bacterial infections in farmed fish [6]. These compounds are widely applied to ensure fish health and productivity, but this use exerts selective pressure on microbial communities, potentially leading to shifts in community structure and the development of antimicrobial resistance [7,8]. Research has shown that antimicrobials use in aquaculture can significantly alter the microbial diversity and composition of surrounding environments, including marine sediments [9]. Evidence suggests that antimicrobial residues and resistant genes may persist in sediments long after treatment end, facilitating horizontal gene transfer among microbial populations [10]. Such exchanges can occur across bacterial species, complicating efforts to manage antimicrobial resistance [11]. Despite these findings, most studies have focused on water-column microbial communities, with less attention given to sedimentary environments beneath and around fish farms [12,13]. This gap is likely due to the challenges of accessing and analyzing sediment samples, given the complexity of the matrix. Notwithstanding, sediments function as both a sink and source of microbial activity and genetic exchange, underscoring the importance of these sediments in understanding the broader ecological impacts of aquaculture.

Chile, the world’s second largest salmon producer after Norway, harvested 998,234 tons in 2024 [14]. The environmental conditions of southern Chile and the extensive use of antimicrobials, primarily florfenicol, in salmon farming [15] highlight the urgent need to better understand the impacts thereof on marine-sediment microbiomes. Florfenicol, a broad-spectrum, semi-synthetic antimicrobial that has seen used intensively and continuously in marine salmon production systems [16], largely due to recurring outbreaks of piscirickettsiosis. This disease has not been effectively controlled through current management practices or vaccinations strategies and accounted for 97% of all antimicrobials used during the marine production stage in 2024 [15].

This study aimed to investigate the composition and diversity of bacterial communities in sediments surrounding farms subjected to antimicrobial treatments, with a focus on identifying shifts in community structure. The findings are expected to significantly advance our understanding of how antimicrobial treatments influence microbial communities in aquaculture-related sediments. Ultimately, this study seeks to improve knowledge on the complex interactions between aquaculture practices and marine-sediment microbiomes, emphasizing the need for integrated management strategies that address both ecological and public health concerns.

## 2. Results

### 2.1. 16S rRNA Sequencing of Marine Sediment Samples

Community structure differences were evaluated across multiple taxonomic levels using 16S rRNA gene amplicon sequencing, complemented by metagenomic analysis and multivariate statistical approaches. Following 16S rRNA sequencing, most samples yielded over 130,000 high-quality reads, with each sample containing between 2000 and 5000 distinct amplicon sequence variants (ASVs) (Figure 1). Rarefaction curves approached saturation, indicating that the sequencing depth was sufficient to capture most of the microbial diversity present in the samples. In terms of taxonomic resolution, the sequencing data, after decontamination, identified 53 phyla, 142 classes, 316 orders, 476 families, and 738 genera.

### 2.2. Shannon Index (H′) Analysis

A progressive increase in Shannon diversity (H′) was observed in bacterial communities from sediments beneath the cages, corresponding to increasing taxonomic resolution across all sites. Mean Shannon Index values rose from ~3.5 at the phylum level, highlighting the higher number of distinguishable ASVs recovered at finer taxonomic ranks. Figure 2 displays alpha diversity metrics across taxonomic levels, as measured by the Shannon Index.

Both antibiotic-treated salmon farms exhibited altered sediment microbiota compared to the untreated Control. Farm 1 showed a significant reduction in Shannon diversity across all taxonomic levels. In contrast, Farm 2 exhibited intermediate diversity, significantly different from Farm 1, but largely similar to the Control, except for a modest yet significant decrease at the class level.

### 2.3. Relative Abundance Analysis

Following ASV assignment, the relative abundance of microorganisms at the phylum level was assessed. Figure 3 illustrates the relative abundance of bacterial ASVs in marine sediment samples from each site. At the control site, Pseudomonadota exhibited the highest relative abundance (up to ~25%), while Bacteroidota and Actinobacteriota were present at moderate levels. In contrast, both Farms 1 and 2 displayed a marked increase in Bacteroidota (up to ~30%) and a consistent decrease in Pseudomonadota. Other phyla such as Fibrobacterota, Bacillota, and Actinobacteriota, remained at low proportions across all samples (<10%).

Comparative analysis between the control and the treated farms revealed pronounced shifts in the relative abundance of several bacterial phyla, including statistically significant differences in Chloroflexi and Firmicutes.

### 2.4. Weighted UniFrac Analysis

Weighted UniFrac analysis revealed a wide dispersion of microbiomes across the three farms at the ASV level (Figure 4).

The first two principal components explained 93.01% of the total variance (PC1: 79.07%, PC2: 7.69%), indicating that most variation in bacterial community composition was captured by PC1.

The plot reveals a clear spatial separation among sample groupings. Replicates from Farm 2 cluster tightly in the upper right quadrant, indicating a relatively homogeneous bacterial community that is distinct from the other groups. Farm 1 samples are positioned on the left side of the plot, clearly separated from both Farm 2 and the control, although they exhibit greater internal variability. In contrast, control samples consistently cluster in the lower central area, clearly differentiated from both treated farms.

### 2.5. Relative Abundance at the Taxonomic Level

Figure 5 presents microbiome comparisons across different taxonomic levels. The control farm exhibited high taxonomic diversity, with a broad representation of phyla such as Proteobacteria, Bacteroidota, Planctomycetota, Chloroflexi, and Desulfobacterota. A substantial portion of taxonomic assignments was classified as “Unclassified” at the lowest taxonomic level. In contrast, Farm 1 showed a noticeable shift in relative abundance, with increased representation of Desulfobacterota and Bacteroidota compared to control site. Notably, orders such as Desulfobulbales and Flavobacteriales were more prominent. Meanwhile, a reduced abundance was observed for phyla such as Planctomycetota and Chloroflexi. The “Unclassified” category remained substantial but showed a more concentrated taxonomic distribution. Farm 2 displayed a profile similar to that of Farm 1, with some distinctions. It exhibited a higher relative abundance of Desulfobacterota, particularly Desulfobulbales, suggesting an enrichment of sulfur-reducing bacteria, with 228 taxonomic assignments associated with this group. Additionally, there was a further decline in the diversity of phyla such as Planctomycetota and Chloroflexi, along with a notable increase in the “Other” category across several taxonomic levels.

The heatmap in Figure 6, which depicts the relative abundances of bacterial families, further highlights the distinct taxonomic profiles associated with each farm. Hierarchical clustering of samples and families revealed clear differences between sites. The Control farm was characterized by higher relative abundances of families such as *Pirellulaceae*, *Nitrosococcaceae* and *Psychromonadaceae*. In contrast, florfenicol-treated sites (Farms 1 and 2) showed enrichment in mainly *Desulfobulbaceae*, *Desulfocapsaceae*, and *Anaerolineaceae*, and unclassified families within Bacteroidota and Firmicutes. Some families, such as *Flavobacteriaceae* and *Woeseiaceae*, were detected at all sites but at varying relative abundances.

### 2.6. Predicted Functional Pathways of Bacterial Communities

Predicted functional pathways revealed distinct metabolic profiles among the farms. In Farm 2, pathways related to L-methionine biosynthesis III and fatty acid oxidation were highly enriched. Farm 1 showed notable enrichment in the glycogen biosynthesis pathway. In contrast, the Control farm exhibited enrichment in several metabolic pathways, including asparagine biosynthesis (ASPASN_PWY) and urea cycle and nitrogen metabolism (URSIN_PWY), among others.

## 3. Discussion

The specific effects of antimicrobial treatments on microbial communities in marine sediments associated with salmon farms remain insufficiently characterized. Most previous studies have focused on culturable bacteria [17], freshwater systems [18,19], or the water-column microbiome [20], leaving a critical knowledge gap regarding sediment-associated microbial ecosystems. This study aimed to address that gap by investigating the effects of florfenicol, a commonly used antimicrobial agent in salmon aquaculture [16], on the composition and diversity of bacterial communities in marine sediments.

To this end, sediment samples were collected from three sites: Farms 1 and 2 (both treated with florfenicol) and a control site (no antimicrobial treatment during the production cycle). Metagenomic analysis provided a comprehensive profile of microbial community structure across these sites.

A progressive increase in Shannon diversity from phylum to species level was observed in all samples, reflecting the expected rise in diversity as taxonomic resolution increases. This trend highlights the importance of fine-scale taxonomic classification, as deeper profiling can reveal environmental heterogeneity that may be overlooked at broader taxonomic levels. Studies limited to higher taxonomic ranks may therefore underestimate microbial diversity loss due to anthropogenic stressors [21].

Florfenicol treatment was associated with reduced microbial diversity, particularly in Farm 1, which showed a consistent decrease in H′ across taxonomic levels. This reduction suggests a broad ecological impact of the antibiotic, likely favoring a limited number of resistant taxa and reducing both richness and evenness [9]. In contrast, Farm 2 exhibited intermediate diversity, more closely resembling the Control site. This may be attributed to differences in antibiotic dosage, exposure duration, or local environmental conditions that influence compound degradation and dispersion.

The marked diversity loss in Farm 1 may serve as an early indicator of ecological disturbance, with potential downstream effects on sediment biogeochemistry, ecosystem functions, and the spread of antibiotic resistance. Long-term monitoring could help determine the persistence of these effects and the recovery potential of benthic microbial communities after antimicrobial exposure. These findings align with prior studies in aquatic environments, which have demonstrated that antibiotic use in aquaculture can reduce microbial diversity, alter community composition, and promote the spread of resistance genes [22,23]. Such changes were evident in this study through shifts in the relative abundance of bacterial taxa at the phylum level.

Specifically, the observed reduction in Pseudomonadota in florfenicol-treated sites (Farms 1 and 2) suggests that many members of this phylum, often aerobic and metabolically versatile, may be particularly susceptible to florfenicol or lack resistance mechanisms [19,24,25]. Conversely, the increased abundance of Bacteroidota, frequently associated with organic-rich environments and known for resistance potential, implies selection for more tolerant or resistant taxa [26]. This pattern aligns with previous findings showing that antibiotic exposure in aquaculture systems can selectively enrich specific bacterial phyla, depending on ecological traits and resistance profiles [27]. Additionally, the substantial presence of the “Other” category in the control farm suggests that florfenicol may also influence less abundant or rare lineages, potentially favoring taxa with adaptive traits (Figure 3). These findings underscore the need for detailed taxonomic and functional investigations, including resistance-gene profiling, to better understand the ecological consequences of antimicrobial use in aquaculture and the role of antimicrobials in shaping sediment microbiota.

The selective pressure exerted by antimicrobials likely promote the proliferation of resistant bacterial taxa while suppressing susceptible populations. This effect may arise from the inherent ability of certain bacterial phyla to harbor and disseminate resistance genes, conferring a survival advantage under antimicrobial stress [28]. Alternatively, florfenicol exposure may disrupt microbial interactions and nutrient cycling within the sediment, resulting in shifts in both community composition and ecosystem function [29].

Weighted UniFrac analysis revealed a substantial dispersion of microbiomes across the three farms, indicating that florfenicol treatment alters the composition of sediment microbial communities. The observed similarity between Farms 1 and 2 suggests that, despite geographic separation, antibiotic exposure exerts a convergent effect on benthic microbial assemblages. In comparison to the control site, treated sites exhibited a clear and consistent separation, supporting the hypothesis that antimicrobial agents act as strong selective pressures, reshaping the relative abundance and phylogenetic diversity of microbial taxa.

The lower dispersion observed in Farm 2 suggests a more homogeneous community, potentially dominated by taxa resistant or tolerant to florfenicol. This is consistent with the hypothesis that antimicrobial selective pressure reduces diversity and promotes the expansion of specific populations with adaptive advantages. In contrast, the greater dispersion observed in Farm 1 may reflect a transitional state or lower selective pressure, allowing for a higher degree of community heterogeneity.

In contrast, microbial communities from Farms 1 and 2 differed markedly from those of the control site, despite a few exceptions among replicates, likely due to the selection of resistant or tolerant microorganisms under antimicrobial pressure. The fact that PC1 accounted for nearly 79.07% of the total variance, with a clear separation between groups, suggests that antibiotic exposure is the primary factor shaping the structure of these microbial communities. This analysis confirmed that florfenicol exposure significantly alters the composition and phylogenetic structure of bacterial communities in marine sediments beneath salmon farms. These findings reinforce growing concerns about the ecological consequences of antimicrobial use in aquaculture, indicating that such treatments may disrupt local biogeochemical processes, promote the persistence and spread of resistance genes, and ultimately reduce overall ecosystem resilience.

Comparative analysis for the relative abundance of the microbiome across the taxonomic level revealed high phylum-level diversity for the control farm. A substantial portion of the taxonomic flow ended in the “Unclassified” category, suggesting the presence of numerous unidentified taxa at finer taxonomic ranks. In contrast, Farm 1 displayed a marked shift in community composition, with an increased representation of groups potentially associated with environmental disturbances or antimicrobial resistance. Concurrently, a reduction in the relative abundance of phyla such as Planctomycetota and Chloroflexi was observed. While the “Unclassified” category remained significant, it showed a more concentrated taxonomic flow, indicating a narrowing of microbial diversity. Farm 2 displayed a community profile similar to that of Farm 1, but with a notable increase in the “Other” category across several taxonomic levels, suggesting a rise in low abundance but detectable taxa that may reflect shifts in rare or opportunistic microbial populations.

Overall, the Control site exhibited a more evenly distributed diversity across multiple phyla and taxonomic levels, consistent with a balanced microbial community under natural, undisturbed conditions [30]. In contrast, both Farm 1 and Farm 2, treated with florfenicol, displayed reduced phylogenetic diversity and a notable enrichment of bacterial groups potentially resistant or adapted to antibiotic stress (e.g., Desulfobacterota, Bacteroidota, Flavobacteriales) [31,32].

The microbial communities in treated farms were increasingly dominated by fewer taxa, with pronounced taxonomic flows toward groups such as Desulfobulbales and Sulfurovaceae, likely reflecting the selective pressure imposed by florfenicol. Relative abundance analysis at the family level (Figure 6) revealed marked shifts in sediment microbial composition, suggesting that florfenicol use significantly influences bacterial community composition in aquaculture sites. Treated farms showed increased relative abundance of anaerobic or facultative anaerobic taxa, including Desulfobulbaceae and Anaerolineaceae, which are commonly associated with sulfidogenic conditions and organic enrichment [33,34]. These taxa may serve as potential biomarkers of environmental changes induced by aquaculture in marine sediments [35]. This pattern suggests that florfenicol application may indirectly favor bacteria adapted to reduced redox conditions, either through direct antimicrobial selection or via florfenicol-mediated alterations in sediment biogeochemistry [36]. Additionally, in the farms treated with florfenicol, a predominance of the *Flavobacteriaceae* family was observed. This group is characterized by several mechanisms to evade antimicrobial action, including, though less frequently, efflux pumps mediated by the specific *floR* gene [37]. In contrast, the control site exhibited a greater representation of nitrifying families such as Nitrosococcaceae and oligotrophic taxa, indicating a more stable and functionally diverse microbial community [38,39].

Finally, predicted functional pathways revealed a notable enrichment of L-methionine biosynthesis III in Farm 2, indicating a shift in sulfur metabolism potentially driven by antimicrobial treatment. Additionally, increased activity in fatty acid oxidation pathways suggests a higher microbial potential for lipid degradation at this site. The L-methionine biosynthesis III pathway, part of the trans-sulphuration route, plays a key role in the microbial sulfur cycle by enabling bacteria to convert sulfur-containing compounds into methionine [40]. In sediments enriched with organic matter from aquaculture activities (e.g., uneaten feed, feces), microbial communities may upregulate sulfur metabolic pathways to cope with elevated concentrations of sulfates, sulfides, and other sulfur species [41]. This metabolic adaptation facilitates anaerobic degradation of organic matter and influences sediment biogeochemistry.

Salmon farming appears to promote the proliferation of bacterial taxa involved in sulfur metabolism, such as Desulfovibrio (sulfate-reducing bacteria) or Methylophaga (methylotrophs). The enrichment of the L-methionine biosynthesis III pathway may reflect a microbial shift towards more efficient sulfur assimilation, enhancing community adaptability in response to organic enrichment from aquaculture inputs [42]. Moreover, this pathway is linked to S-adenosylmethionine production, a critical molecule involved in DNA methylation and horizontal gene transfer [43]. Therefore, enhanced sulfur metabolism may indirectly promote the spread of antibiotic resistance genes by facilitating methylation-dependent gene expression and transfer mechanisms [44].

In Farm 1, the enrichment of the glycogen biosynthesis pathway may reflect energy adaptations by bacterial communities in response to stress [45]. In contrast, the Control farm showed enrichment in several metabolic pathways, suggesting that bacterial communities in the absence of antimicrobial inputs maintain a distinct and potentially balanced metabolic profile [46].

The observed increase in sulfur metabolism (e.g., methionine biosynthesis III) and lipid degradation in Farm 2 may indicate microbial responses to organic overloading. Meanwhile, the increased energy storage pathways in Farm 1 suggest a different mode of microbial adaptation to environmental stressors. In contrast, the Control sediments retained more diverse and specialized metabolic functions, likely reflecting less disturbed environmental conditions, despite the basal impact inherent in farmed salmon production.

The presented findings reinforce the notion that antimicrobial—specially florfenicol—inputs in aquaculture not only affect microbial taxonomic composition but also reshape key biogeochemical functions, including nitrogen and sulfur cycling, potentially compromising benthic ecosystem resilience. In this context, further analysis of the functional roles of enriched microbial families is essential to better understand the ecological consequences of aquaculture practices in marine environments. Future studies incorporating longer time scales and additional site comparisons are necessary to deepen our understanding of how antimicrobial treatments in salmon farming influence sediment microbial communities. Furthermore, integrating resistome-focused approaches as a complementary strategy would enable direct assessment of whether the observed compositional shifts translate into the selection, enrichment, or mobilization of resistance determinants.

## 4. Materials and Methods

### 4.1. Study Design

This study was designed to evaluate the impact of florfenicol treatments on bacterial communities in marine sediments from Atlantic salmon (*Salmo salar*) farms in Chile. The experimental design included three salmon farms located in the Los Lagos Region of southern Chile, selected based on a documented history of antimicrobial use. Farms 1 and 2 were designated as experimental groups; in both cases, fish with an average weight of 3990 g and 5200 g, respectively, were diagnosed with outbreaks of piscirickettsiosis. As a result, a veterinarian prescribed oral florfenicol treatments lasting 33 days for Farm 1 and 20 days for Farm 2, with a dose of 17 mg kg^−1^ body weight per day. A third farm served as the control group, as it did not receive any antimicrobial treatment throughout the entire production cycle. This setup enabled a comparative analysis of sediment-associated microbial communities between treated and untreated farms.

### 4.2. Collection of Sediment Samples from Salmon Farms

Marine sediment samples were collected from each farm using a van Veen dredge to ensure consistent sampling [46]. Sampling was conducted at three distinct locations within each farm, with each location sampled in triplicate to account for spatial variability. Samples were taken from the top 10 cm of the sediment surface.

For Farm 1, sediment samples were collected directly beneath the cage at a depth of 60 m, 15 m from the cages at a depth of 40 m, and 80 m from the cages at a depth of 30 m. For Farm 2, samples were collected beneath the cage at a depth of 40 m, 15 m from the cages at a depth of 30 m, and 100 m from the cages, also at a depth of 30 m. For the Control farm, sediment samples were collected beneath the site cage at a depth of 50 m, 15 m from the cage at 40 m depth, and 100 m from the cage, also at 40 m depth. Each sample was placed in sterile polyethylene bags and transported at 4 °C to the laboratory for immediate processing to preserve microbial integrity.

### 4.3. Total DNA Extraction

Total DNA was extracted from all sediment samples using the DNeasy PowerSoil Pro Kit (Qiagen, Hilden, Germany) following the manufacturer’s instructions. DNA purity was assessed spectrophotometrically using an Epoch Take 3 (BIOTEK, Winooski, VT, USA) microplate reader by measuring the absorbance ratio at 260/280 nm. The concentration of double-stranded DNA was quantified using a Qubit 3.0 fluorometer with the Qubit dsDNA High Sensitivity Assay Kit (ThermoFisher, Waltham, MA, USA). Genomic DNA integrity was evaluated by 1% agarose gel electrophoresis (80 V, 40 min) and further verified using either capillary electrophoresis on an Agilent Fragment Analyzer (AATI) or automated electrophoresis with a Bioanalyzer TapeStation 4150 (Agilent Technologies, Santa Clara, CA, USA), depending on the DNA integrity number of each sample.

Subsequently, each sample was precipitated with three volumes of cold ethanol (Merck, EMSURE, Darmstadt, Germany) and incubated at −20 °C for 2–3 h. The samples were then centrifuged at 14,000× *g* for 15 min at 4 °C. After discarding the supernatant, the resulting pellets were washed with 500 μL of 70% ethanol. DNA was then resuspended in the TE buffer (10 mM Tris pH 8.0, 1 mM EDTA). DNA concentrations were quantified using spectrophotometry and fluorimetry, as described previously.

The resulting DNA was subjected to electrophoresis on low-melting point agarose gels (Invitrogen, Carlsbad, CA, USA). Genomic DNA products were excised and purified using the Wizard^®^ SV Gel and PCR Clean-Up System (Promega, Madison, WI, USA). Purified products were evaluated by spectrophotometry, fluorimetry, and agarose gel electrophoresis, as described previously.

### 4.4. Preparation of 16S rDNA Sequencing Libraries and Bioinformatic Analysis

To amplify the V3-V4 region (300 bp) of the 16S rRNA gene, DNA samples were subjected to polymerase chain reaction (PCR) using primers 341F (5′-CCTACGGGNGGCWGCAG) and 806R (5′-GGACTACNVGGGTWTCTAAT-3′). Illumina sequencing adapters and dual-index barcodes were added to the amplicons. Sequencing was performed at the Centro Bioinformática de Sistemas of the Universidad Andrés Bello (Santiago, Chile) using the Illumina MiSeq Platform (San Diego, CA, USA).

For the comparative bioinformatic analysis between Farms 1 and 2 and the Control group, the three sampling stations per farm were treated as biological triplicates. Sequencing data were processed using a bioinformatics pipeline comprising the following steps: (1) sample demultiplexing using QIIME v1.8.0 [47]; (2) sequence denoising and taxonomic assignment with DADA2 v1.22 [48]; (3) metabolic function prediction with (PICRUSt2) [49]; and (4) statistical analyses using scipy v1.10.0 [50] and LEfSe (Linear Discriminant Analysis Effect Size [51]) for biomarker discovery.

Individual samples were extracted from the FASTQ file using the QIIME demultiplex_fasta.py script. DADA2 was then used to remove low-quality reads, trim sequences to 220 nucleotides, and filter out reads with ambiguous bases. A sequencing error model was inferred and used to generate ASVs.

The taxonomic assignment of ASVs was performed using a Naïve Bayesian classifier [52] against the SILVA database v138 [53,54], employing the assign taxonomy function in DADA2 with default parameters (minimum confidence = 50%). ASVs with a relative abundance below 0.1% were excluded from downstream analysis.

The resulting ASV abundance table was used to predict metabolic functions and pathways via PICRUSt2, referencing the MetaCyc database. The rarefaction curve was determined using the vegan package v2.6-4 (https://cran.r-project.org/web/packages/vegan/index.html, accessed on 20 August 2025). Alpha diversity (Shannon Index) was calculated at multiple taxonomic levels using the skbio.diversity.alpha.shannon function from the scikit-bio-Python package.

### 4.5. Statistical Analyses

The Mann–Whitney U-test from the Python SciPy package v1.10.0 [50] was used to analyze differences in the relative abundance of ASVs at various taxonomic levels between the Control and florfenicol-treated farm groups. Additionally, the LEfSe [51] was applied to identify statistically significant differences in the relative abundance of predicted metabolic functions and pathways among the farms. For LEfSe, data were normalized to one million reads per sample prior to analysis. Statistical significance was defined as *p*-value < 0.05 for the Mann–Whitney U test and a log_10_ LDA score ≥ 2.0 for LEfSe.

To assess beta diversity, the weighted UniFrac method was applied to the relative abundance data at the phylum level using the scikit-bio-Python package v0.5.8 (http://scikit-bio.org). Principal component analysis was performed on the predicted metabolic function and pathway abundance data using scikit-learn v1.1.2 [55]. Pearson correlation coefficients were calculated using the pearsonr function from SciPy v1.14.1 [50].

## 5. Conclusions

Overall, the presented findings provide important insights into the impact of antimicrobial treatment on the composition, diversity, and functional potential of bacterial communities in marine sediments associated with salmon aquaculture. Florfenicol treatment significantly altered the microbial community structure, leading to a reduction in taxonomic diversity and an enrichment of bacterial taxa potentially resistant or tolerant to antibiotic exposure. In contrast, the untreated Control site exhibited a more diverse and evenly distributed community, characteristic of a stable benthic ecosystem.

Moreover, the presence of specific metabolic pathways, such as those involved in sulfur metabolism and energy storage, suggests that the microbial community is adapting to organic and sulfur-rich conditions induced by aquaculture activity. These functional shifts may affect nutrient cycling, the persistence of antimicrobial resistance genes, and overall sediment health.

These results underscore the ecological consequences of antibiotic use in aquaculture, particularly the selective pressure exerted on benthic microbial communities. They also highlight the value of microbial community profiling as a tool to assess environmental impacts and to identify key microbial indicators for monitoring and mitigation. To ensure the long-term sustainability of salmon aquaculture, it is essential to implement management strategies that reduce antimicrobial use and its environmental footprint. Further research is needed to understand the long-term effects of antimicrobial inputs, to elucidate the mechanisms driving microbial shifts, and to explore alternative disease-control practices that safeguard both aquaculture productivity and ecosystem health.

## Figures and Tables

**Figure 1 antibiotics-14-01016-f001:**
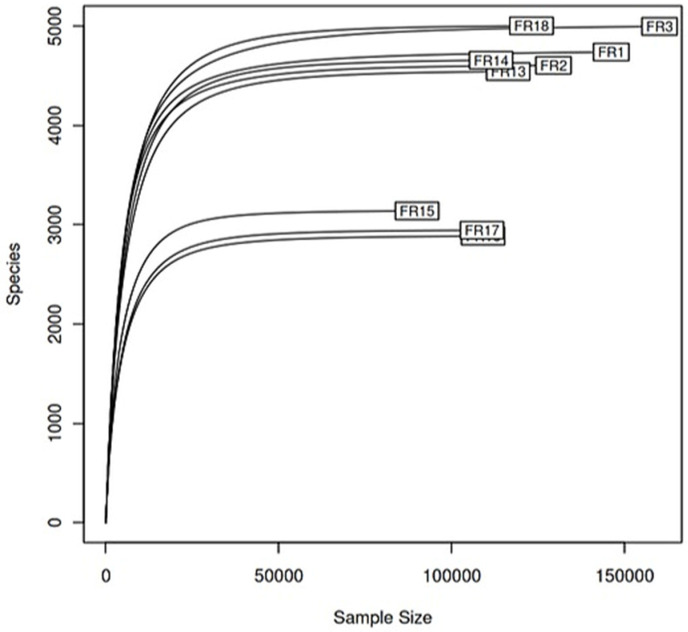
Sequencing statistics. Rarefaction curve showing the number of identified Amplicon Sequence Variants (ASVs) as a function of sequencing depth (x-axis). The y-axis “Species” represents ASV richness across samples. Control farm: FR 1–3; Farm 1: FR 13–15; and Farm 2: FR 16–18.

**Figure 2 antibiotics-14-01016-f002:**
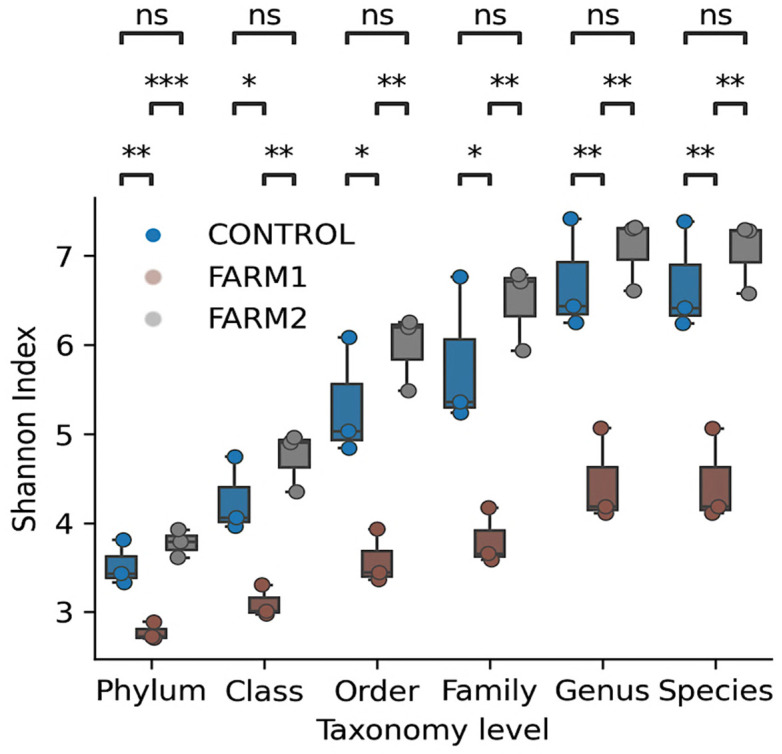
Shannon index (alpha diversity) of total reads across taxonomic levels. Dark blue dots represent Amplicon Sequence Variants (ASVs) from the Control group. Dark brown dots represent ASVs from Farm 1, and dark gray dots represent ASVs from Farm 2. Asterisks indicate statistically significant differences (* *p* < 0.05, ** *p* < 0.01, *** *p* < 0.005).

**Figure 3 antibiotics-14-01016-f003:**
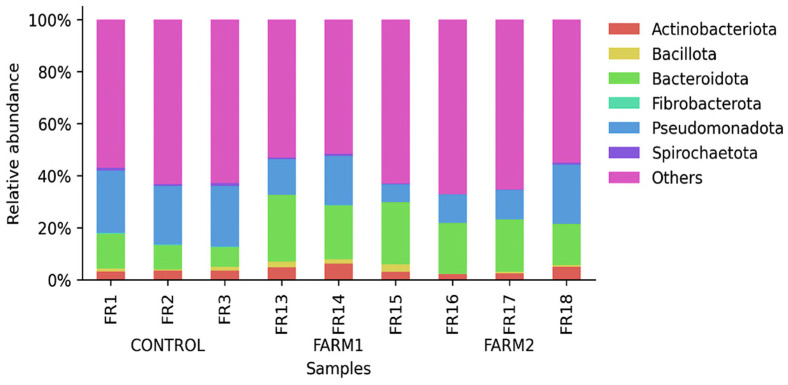
Relative abundance per sample at the phylum level. The “Others” category includes Amplicon Sequence Variants (ASVs) with an average relative abundance of less than 1% within each group.

**Figure 4 antibiotics-14-01016-f004:**
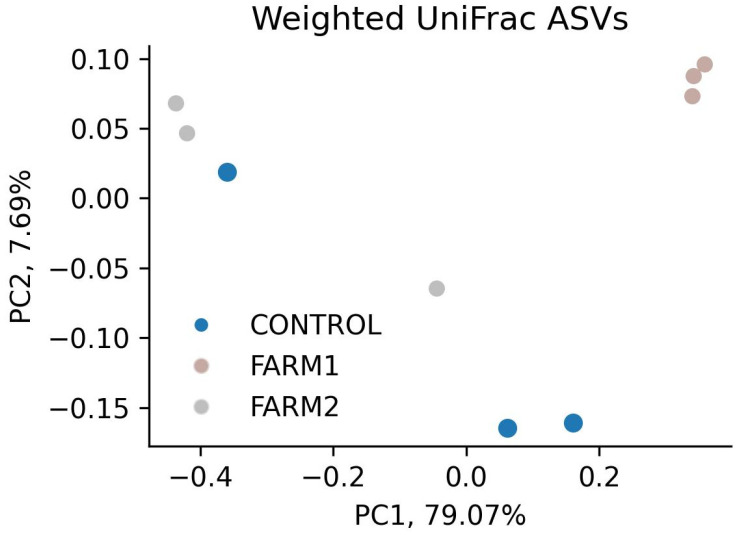
Weighted UniFrac analysis revealed high dispersion among samples from Farms 1 and 2, indicating greater variability in microbial community composition compared to the Control group.

**Figure 5 antibiotics-14-01016-f005:**
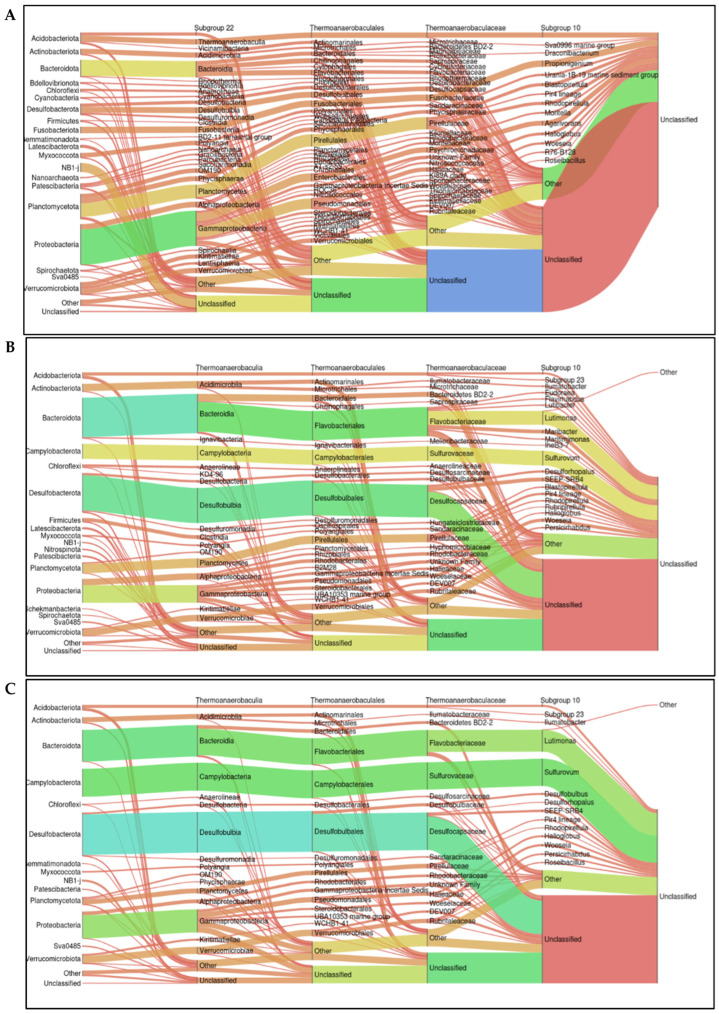
Sankey plots showing the average relative abundance across taxonomic levels for each group: (**A**) Control, (**B**) Farm 1, and (**C**) Farm 2.

**Figure 6 antibiotics-14-01016-f006:**
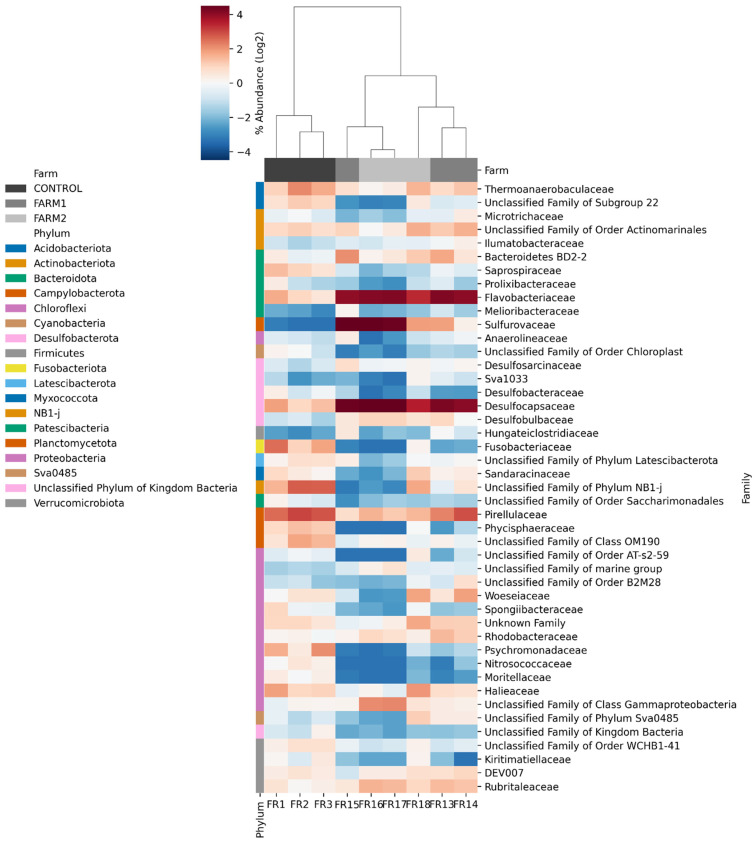
Heatmap of the relative abundance of bacterial families across all samples. Rows represent families with >1% relative abundance in at least one sample. Columns correspond to sediment samples collected from each farm: Control (FR 1–3), Farm 1 (FR 13–15), and Farm 2 (FR 16–18).

## Data Availability

Data are available within the article, and upon request from the corresponding authors. Additionally, the datasets generated and analyzed during this study are accessible in the European Nucleotide Archive repository under the code PRJEB98379 (https://www.ebi.ac.uk/ena).

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
