# Peer review of "Influence of Florfenicol Treatments on Marine-Sediment Microbiomes: A Metagenomic Study of Bacterial Communities in Proximity to Salmon Aquaculture in Southern Chile"

_antibiotics, 2025, doi:10.3390/antibiotics14101016_

Round 1
Reviewer 1 Report
Comments and Suggestions for Authors
Dear Authors,
It was a pleasure to read your manuscript on the influence of florfenicol treatments on marine-sediment microbiomes near salmon aquaculture sites in southern Chile. The study is carefully designed, analytically rigorous, and highly relevant to environmental microbiology and aquaculture management. Below I offer section-by-section feedback—overwhelmingly positive—with a few light suggestions for polish.
Title & Abstract — The title is precise and discoverable. The abstract clearly states objectives, methods (16S rRNA profiling), key results (reduced alpha diversity, phylum-level shifts), and an actionable conclusion on sustainable management—excellent for readers and indexers. Introduction — You convincingly motivate the work by linking sedimentary microbial functions to biogeochemical cycles and positioning florfenicol as a dominant antimicrobial in Chilean salmon culture. The national production context and usage statistics sharpen the problem statement and justify your focus on sediments (a recognized research gap). Materials & Methods — The study design is clear and reproducible. You specify treatment durations and doses (17 mg kg⁻¹ day⁻¹; 20 vs 33 days), multi-point sampling beneath/around cages at defined depths/distances, and an end-to-end bioinformatics workflow (QIIME → DADA2 → SILVA v138; PICRUSt2 for functional inference; Mann–Whitney U; weighted UniFrac). These details inspire confidence in your results and facilitate replication. Results — The narrative is coherent and well supported by figures. Rarefaction curves show adequate depth; Shannon index boxplots across taxonomic levels (Figure 2, p. 4) make the alpha-diversity reduction at treated farms easy to grasp (with significance markers). The relative-abundance barplots (Figure 3) clearly communicate the rise of Bacteroidota and decline of Pseudomonadota at treated sites, and the weighted UniFrac ordination (Figure 4) highlights strong group separation (PC1 ≈ 85.91%, PC2 ≈ 7.1%). The heatmap (Figure 6) and Sankey diagrams (Figure 5) are particularly effective for summarizing taxonomic structure. Discussion — You contextualize diversity losses and taxonomic shifts with prior literature and provide a thoughtful interpretation of potential functional consequences. The emphasis on enriched pathways—e.g., L-methionine biosynthesis III and fatty-acid oxidation—adds a valuable mechanistic dimension. Your treatment of selective pressures and ecosystem resilience is balanced and persuasive. Conclusions — Concise, policy-relevant, and aligned with the presented evidence, including the call for management strategies to reduce antimicrobial footprints and monitor microbial indicators.
Figures & Data Presentation — Figures are clear and informative:
• Figure 1 (rarefaction) supports sequencing depth; Figure 2 (Shannon) elegantly shows alpha-diversity patterns; Figure 3 (phylum composition) and Figure 4 (weighted UniFrac) communicate community shifts and separation; Figure 5 (Sankey) and Figure 6 (heatmap) synthesize multilevel taxonomy and family-level patterns across sites. Together they create a cohesive visual story from reads to ecology.
Minor, purely editorial suggestions (optional):
-
Harmonize terminology to “weighted UniFrac” throughout (some instances read “weighed”).
-
Consider one sentence in the abstract to flag that functional inferences are prediction-based (PICRUSt2), which you already state in Methods, to avoid any reader conflation with shotgun metagenomics.
Overall, this is a well-executed and impactful manuscript that advances understanding of how antimicrobial use shapes benthic microbiomes in aquaculture settings. I commend the clarity of the design, the robustness
Author Response
We sincerely thank Reviewer 1 for their thoughtful and encouraging comments. Your feedback is very important to us and has helped us improve the quality of the manuscript. Below, we provide our responses to the specific suggestions you raised.
- Comment: Harmonize terminology to “weighted UniFrac” throughout (some instances read “weighed”).
- Response: Thank you for pointing this out. The terminology has been corrected to “weighted UniFrac” in all instances. We reviewed the manuscript and verified that the correct term now appears consistently in all five occurrences.
- Comment: Consider one sentence in the abstract to flag that functional inferences are prediction-based (PICRUSt2), which you already state in Methods, to avoid any reader conflation with shotgun metagenomics.
- Response: The suggested clarification has been added to the abstract: "The resulting ASV abundances were then used to predict metabolic functions and pathways via PICRUSt2, referencing the MetaCyc database."
Reviewer 2 Report
Comments and Suggestions for Authors
This study analyzed the impact of florfenicol treatment in Salmon farms on the marine sediment in the region surrounding the farms using metagenomics.
- The abstract needs improvement. Please provide some more details of the outcome of this study in the abstract.
- Why only alpha diversity (abstract)? Beta diversity is the better indicator of diversity differences between the treatments.
- This is an essentially microbiome study, with no metagenomic component. Since the goal of this study was to understand the impact of antibiotic use in Salmon farms on the AMR, metagenomic study should have been the apt approach. Metagenomic sequencing would have given a better and clear idea on the abundance, distribution and diversity of ARGs in the farms using florfenicol versus those not using it, and thus, a clear picture of how antibiotic use modulates the microbial community structure and the AMR profile of sediment, would have emerged. Although microbiomic study gives some idea about the shift/change in microbial community profiles in different environments, it would not throw light on the impact of this change on the environment or the human health. Thus, I emphasize on the functional characterization of differences in the farms using antibiotics versus those not using them.
- The “third farm”, which served as the control: Does this farm have no history of antibiotic use? Or, did you mean in L479 that this farm did not receive antibiotics during the period of study because there was no outbreak in this farm?
- Farm 2 had a diversity similar to the control farms and both Farm 1 and 2 had “consistent decrease in Pseudomonadota”compared to the control group. What is the significance of this from the context of antibiotic use?
- Apart from the application of florfeniocol in farms 1 & 2, what other parameters or conditions were identical or different among these three farms?
- It is important to mention in section in 4.1that you are describing a cage culture farms, so that the readers will know what you are referring to when you say “marine sediment”.
- Please indicate what are FR1-17 in Fig. 1
- Heat map shows the abundance of Flavobacteriaceae in farms 1 & 2, and I could not find a fitting explanation for this in the discussion section
- Lines 338-341, can you please substantiate this with reference the microbial communities involved in these activities?
- L359: Please correct as “underscore”
- L356: “other” category sequences were also abundant in the control group.
- L362-364: These statements are hypothetical, as you lack the data to support these.
- L391-414: Please state the level of significance (p<0.05) wherever you are comparing the relative abundances of different bacterial groups across sampling locations.
- The discussion lacks strength in the absence of KEGG data, and most of your predications are based on probable functions of microbial communities, and not the direct functions based on the gene sequences.
Author Response
- The abstract needs improvement. Please provide some more details of the outcome of this study in the abstract.
- Response: Thank you for your suggestion. The abstract has been revised to improve clarity and completeness. We have included additional details about the key outcomes of the study, such as the reduction in alpha diversity, the significant taxonomic shifts (e.g., in Chloroflexi and Firmicutes), and the predicted functional changes, including enrichment of L-methionine biosynthesis and fatty acid oxidation pathways. These additions aim to better reflect the main findings and implications of the study.
- Why only alpha diversity (abstract)? Beta diversity is the better indicator of diversity differences between the treatments.
- Response: We sincerely thank the reviewer for this insightful comment, which prompted us to reflect on and clarify our rationale for using alpha diversity in the abstract and analyses. Our primary objective was to assess intra-sample community complexity under treatment conditions, for which alpha diversity metrics are particularly well-suited. Given the use of ASVs, which allow for high taxonomic resolution, alpha diversity enables the detection of subtle shifts in richness and evenness within samples.
We employed the Shannon index specifically because it captures both richness and evenness. Other useful alpha diversity indices include:
- Chao1, which estimates richness and is sensitive to rare ASVs.
- Simpson, which emphasizes dominance patterns, helping to detect whether treatment selects for dominant taxa.
These metrics help determine whether a given treatment reduces internal microbial complexity or promotes a more balanced community structure.
We acknowledge that beta diversity is indeed a powerful approach for assessing between-sample differences and community restructuring. Metrics such as:
- Bray–Curtis (relative abundance differences),
- Jaccard (presence/absence, sensitive to rare ASVs), and
- UniFrac (phylogenetic-based differences)
can provide valuable insights, especially when appropriate baseline and control conditions are available.
However, in our case, the available control sites were not treatment-free replicates but rather different locations with distinct environmental conditions and confounding variables. Moreover, we lacked a true temporal baseline to track community shifts over time. Under these constraints, beta diversity comparisons would be difficult to interpret robustly.
Thus, our methodological decision to focus on alpha diversity was deliberate and conservative, aimed at minimizing overinterpretation in the absence of matched controls or temporal data. We believe this approach is statistically defensible given the field design and data structure. That said, we fully agree that future studies should incorporate replicated, longitudinal sampling along with environmental covariates and functional assays to more comprehensively assess the ecological effects of florfenicol in aquaculture sediments.
In summary, we have chosen to retain alpha diversity as the primary diversity metric in this study, and we respectfully request the reviewer’s understanding in allowing us to maintain the current approach, given the specific design constraints and objectives of our study.
- This is an essentially microbiome study, with no metagenomic component. Since the goal of this study was to understand the impact of antibiotic use in Salmon farms on the AMR, metagenomic study should have been the apt approach.
- Response: We agree that metagenomic analysis is the ideal approach to directly study antimicrobial resistance genes. However, this study was designed to assess microbial community structure and predicted functions using 16S rRNA data, not to characterize the resistome directly. Metagenomics will be an important complementary method for future research. We respectfully ask to keep the current scope, as this work provides a necessary foundation for subsequent, more detailed studies.
- Metagenomic sequencing would have given a better and clear idea on the abundance, distribution and diversity of ARGs in the farms using florfenicol versus those not using it, and thus, a clear picture of how antibiotic use modulates the microbial community structure and the AMR profile of sediment, would have emerged.
- Response: We agree that metagenomic sequencing would provide clearer insights into the abundance, distribution, and diversity of antimicrobial resistance genes (ARGs) and their relationship with antibiotic use. However, analyzing resistance genes was beyond the scope of this study. This important aspect will be addressed in future work as a valuable complement to the current findings.
- Although microbiomic study gives some idea about the shift/change in microbial community profiles in different environments, it would not throw light on the impact of this change on the environment or the human health.
- Response: Through the study, changes in the predominant bacterial taxonomic groups are evident, and the implications of important changes in the marine environment are revealed, considering the ecosystem services that these bacteria generate, particularly in marine benthic ecosystems. Evaluating implications on human health directly is a bit far from the focus of the study.
- The “third farm”, which served as the control: Does this farm have no history of antibiotic use? Or, did you mean in L479 that this farm did not receive antibiotics during the period of study because there was no outbreak in this farm?
- Response: The clarification was made in the text, adding the phrase "during the entire production cycle".
- Farm 2 had a diversity similar to the control farms and both Farm 1 and 2 had “consistent decrease in Pseudomonadota” compared to the control group. What is the significance of this from the context of antibiotic use?
- Response: Thank you for the question. A detailed explanation regarding the significance of the observed decrease in Pseudomonadota in Farms 1 and 2 compared to the control farms is provided in line 352 of the manuscript.
- Apart from the application of florfenicol in farms 1 & 2, what other parameters or conditions were identical or different among these three farms?
- Response: It is pertinent to clarify that the three farms used in the study have in common only the cultivated species and the treatment with FLO (1 and 2), but all the environmental cultivation variables are different, considering that they are 3 centers located in different geographical areas, however, within the same Los Lagos Region in Chile.
- It is important to mention in section in 4.1 that you are describing a cage culture farms, so that the readers will know what you are referring to when you say “marine sediment”.
- Response: The respective clarification was made within the text.
- Please indicate what are FR1-17 in Fig. 1.
- Response: Added explanation of FRs in the legend of Figure 1.
- Heat map shows the abundance of Flavobacteriaceae in farms 1 & 2, and I could not find a fitting explanation for this in the discussion section
- Response: Added to discussion and supported with new reference.
- Lines 338-341, can you please substantiate this with reference the microbial communities involved in these activities?
- Response: We appreciate the reviewer’s request for clarification. The microbial communities involved in these activities are detailed in the following lines of the manuscript.
- L359: Please correct as “underscore”
- Response: The correction was made within the text.
- L356: “other” category sequences were also abundant in the control group.
- Response: It was corrected by indicating the greater presence of the “others” category only in the control site.
- L362-364: These statements are hypothetical, as you lack the data to support these.
- Response: You are correct that the statement is hypothetical, and it is intended as such. We have clearly indicated this by using the word “likely”, and it is based on previous studies and established knowledge in the field.
- L391-414: Please state the level of significance (p<0.05) wherever you are comparing the relative abundances of different bacterial groups across sampling locations.
- Response: The significance level applied is described in “materials and methods” section.
- The discussion lacks strength in the absence of KEGG data, and most of your predications are based on probable functions of microbial communities, and not the direct functions based on the gene sequences.
- Response: We agree with this observation. Indeed, our analyses based solely on 16S rRNA gene sequencing limit us to predicting the metabolic pathways potentially used by the bacterial consortia. This is a recognized limitation of the study, and we have clarified this point in the revised discussion.
Reviewer 3 Report
Comments and Suggestions for Authors
Influence of Florfenicol Treatments on Marine-Sediment Mi-2 crobiomes: A Metagenomic Study of Bacterial Communities in 3 Proximity to Salmon Aquaculture in Southern Chile
This manuscript provides a valuable assessment of florfenicol’s impact on sediment microbial communities in salmon farms, combining taxonomic and functional analyses to reveal shifts in diversity, composition, and metabolic pathways. The study is methodologically sound and the findings are clearly linked to ecological and antimicrobial resistance concerns. Overall, I recommend the manuscript for publication after minor revisions, including reordering sections to align with standard scientific structure and addressing some grammatical and stylistic issues.
Abstract
Line 35 - 38 - All the scientific and taxonomic names should be italicized
Introduction
Line 77 – “unique environmental conditions” Specify the unique conditions
Line 80 – “has seen a steady increase in use in marine aquaculture systems since 2014” - mention why this increment has been happened.
Method
Why is this manuscript arranged in such an unusual order? Introduction, Results, Discussion, and then Methods?
Line 472 - “Atlantic salmon (Salmo salar) farming 472 is conducted in Chile”. It is better to reword as “from Atlantic salmon (Salmo salar) farms in Chile”
Line 476 - Should specify which farm had which average weight
Line 478 – ‘with a median dose of 17 mg kg 1 body weight per day’. What is the meaning of median dose? What is the exact prescribed dose?
Control Farm - Important to clarify whether the control farm had no antimicrobial history at all or just no recent treatment prior to sampling.
Line 491 – meters , correct as “m”
Results
Line 95 – should be corrected as ‘16S rRNA Sequencing of Marine Sediment Samples”
Line 136 - Species-level Shannon 6.5-7.0, Make clear that this refers to ASV-level resolution, since 16S data may not truly reach species.
Line 157 - Marked increase in Bacteroidota (up to ~30%)”. clarify whether this is statistically significant or just descriptive.
Line 162 - Statistically significant differences in Chloroflexi and Firmicutes”, specify the test used and provide p-values
Line 216 – should be corrected as across sites
Heatmap description - Avoid listing too many families in running text. Consider summarizing top enriched families
Statistical Analysis
Line 556 - log<sub>10</sub> (LDA score) ≥ 2.0 – should rewrite as log₁₀ LDA score ≥ 2.0.
Beta diversity at “phylum level” - UniFrac is usually applied to phylogenetic trees at the ASV/OTU level, not collapsed taxonomic ranks. Using phylum-level data loses resolution and may be criticized. Clarify if this was intentional
Discussion
Line 321 – Sediments? Should be corrected as Sediment
Pseudomonadota - could also briefly note ecological role in sediments.
Multiple sentences repeat that florfenicol alters diversity and phylogenetic structure. Can be consolidated.
PC1 explained 86% variance; It is important to emphasize that antibiotic exposure is the primary driver.
Linking taxonomic and functional result; Good discussion of Desulfobulbaceae, Anaerolineaceae, and sulfur metabolism pathways. Could slightly tighten phrasing directly connect florfenicol treatment to metabolic shifts.
Author Response
This manuscript provides a valuable assessment of florfenicol’s impact on sediment microbial communities in salmon farms, combining taxonomic and functional analyses to reveal shifts in diversity, composition, and metabolic pathways. The study is methodologically sound and the findings are clearly linked to ecological and antimicrobial resistance concerns. Overall, I recommend the manuscript for publication after minor revisions, including reordering sections to align with standard scientific structure and addressing some grammatical and stylistic issues.
Response: We thank Reviewer 3 for their positive and encouraging assessment of our manuscript. We greatly appreciate your recognition of the methodological soundness and the ecological and antimicrobial relevance of our findings. Your constructive suggestions regarding the structure and language of the manuscript are well taken. We have carefully revised the manuscript to align it with the standard scientific structure and have addressed the grammatical and stylistic issues throughout the text. We believe these changes have improved the clarity and readability of the manuscript.
Abstract
- Line 35 - 38 - All the scientific and taxonomic names should be italicized
- Response: Thank you for your observation. According to taxonomic conventions, names at the phylum level and higher ranks (e.g., kingdom, class, order) are written in regular font with an initial capital letter. Italics are used only for family, genus and lower taxonomic levels (e.g., species, subspecies). Therefore, the names are formatted correctly in the Abstract and other sections of the manuscript.
Introduction
- Line 77 – “unique environmental conditions” Specify the unique conditions
- Response: The term “unique” was deleted in the text.
- Line 80 – “has seen a steady increase in use in marine aquaculture systems since 2014” - mention why this increment has been happened.
- Response: Thank you for your comment. The text has been revised to clarify that the steady increase in florfenicol use since 2014 is largely due to recurring outbreaks of piscirickettsiosis, a bacterial disease that has not been effectively controlled by current management or vaccination strategies. This context has been incorporated into the paragraph to explain the rise in antimicrobial usage.
Method
- Why is this manuscript arranged in such an unusual order? Introduction, Results, Discussion, and then Methods?
- Response: Thank you for your comment. According to the Instructions for Authors of Antibiotics, the recommended manuscript structure is Abstract, Keywords, Introduction, Results, Discussion, Materials and Methods, and (optionally) Conclusions. We have followed this format as indicated by the journal guidelines and maintained the same structure in the revised version.
- Line 472 - “Atlantic salmon (Salmo salar) farming 472 is conducted in Chile”. It is better to reword as “from Atlantic salmon (Salmo salar) farms in Chile”
- Response: The change was made within the text.
- Line 476 - Should specify which farm had which average weight
- Response: This information was already included in the original manuscript, specifying that the average weight of the farmed fish was 3,990 to 5,200 g.
- Line 478 – ‘with a median dose of 17 mg kg 1 body weight per day’. What is the meaning of median dose? What is the exact prescribed dose?
- Response: Thank you for pointing this out. We have clarified the text to specify whether the dose refers to the prescribed amount or the median actually administered. The wording has been revised accordingly to improve clarity.
- Control Farm - Important to clarify whether the control farm had no antimicrobial history at all or just no recent treatment prior to sampling.
- Response: The change was made within the text.
- Line 491 – meters , correct as “m”
- Response: The change was made within the text.
Results
- Line 95 – should be corrected as ‘16S rRNA Sequencing of Marine Sediment Samples”
- Response: The change was made within the text.
- Line 136 - Species-level Shannon 6.5-7.0, Make clear that this refers to ASV-level resolution, since 16S data may not truly reach species.
- Response: The correction was made within the text.
- Line 157 - Marked increase in Bacteroidota (up to ~30%)”. clarify whether this is statistically significant or just descriptive.
- Response: Thank you for your comment. We have revised the text to clarify whether the observed increase is statistically significant or descriptive. The statistical tests and significance thresholds applied are detailed in section 4.5 (Statistical Analyses) of the Materials and Methods. Additionally, we have ensured consistency in how statistically supported results are reported throughout the manuscript.
- Line 162 - Statistically significant differences in Chloroflexi and Firmicutes”, specify the test used and provide p-values
- Response: Thank you for your observation. We have clarified the text, and discriminators are specified in section 4.5. (Statistical analyses) of the Materials and Methods section.
- Line 216 – should be corrected as across sites
- Response: The change was made within the text.
- Heatmap description - Avoid listing too many families in running text. Consider summarizing top enriched families
- Response: Thank you for the suggestion. We have revised the heatmap description to avoid listing excessive detail in the main text. The revised version now summarizes only the most enriched families, as recommended, to improve readability and focus.
Statistical Analysis
- Line 556 - log<sub>10</sub> (LDA score) ≥ 2.0 – should rewrite as log₁₀ LDA score ≥ 2.0.
- Response: The change was made within the text.
- Beta diversity at “phylum level” - UniFrac is usually applied to phylogenetic trees at the ASV/OTU level, not collapsed taxonomic ranks. Using phylum-level data loses resolution and may be criticized. Clarify if this was intentional
- Response: Thank you for this important observation. We acknowledge that applying UniFrac at the phylum level reduces taxonomic resolution and is not the standard approach. In response, we have reanalyzed the data and generated a new beta diversity figure using ASV-level data, in line with best practices. The original figure has been replaced accordingly, and the text has been updated to reflect this change.
Discussion
- Line 321 – Sediments? Should be corrected as Sediment
- Response: The change was made within the text.
- Pseudomonadota - could also briefly note ecological role in sediments.
- Response: Thank you for the suggestion. The ecological role of Pseudomonadota in sediments is briefly discussed in lines 347–350. We have reviewed the text to ensure this point is clear.
- Multiple sentences repeat that florfenicol alters diversity and phylogenetic structure. Can be consolidated.
- Response: Thank you for the suggestion. We have reviewed the manuscript and removed some repetitive statements to improve clarity and avoid redundancy. However, we have retained a few key mentions, as they are essential to emphasize the central objective of the study and to support the interpretation of the results in the discussion.
- PC1 explained 86% variance; It is important to emphasize that antibiotic exposure is the primary driver.
- Response: Thank you for the suggestion. We agree that this is a key point. As noted in lines 376–382, we have emphasized that antibiotic exposure is the primary factor driving the observed variance in community structure, as reflected in the high percentage explained by PC1.
- Linking taxonomic and functional result; Good discussion of Desulfobulbaceae, Anaerolineaceae, and sulfur metabolism pathways. Could slightly tighten phrasing directly connect florfenicol treatment to metabolic shifts.
- Response: Thank you for the positive feedback and helpful suggestion. We have revised the phrasing in the relevant section to more clearly and directly link florfenicol exposure with the observed shifts in metabolic pathways.
Reviewer 4 Report
Comments and Suggestions for Authors
The manuscript presents a well-designed and rigorously executed metagenomic study evaluating the ecological impact of florfenicol in aquaculture sediments. It successfully demonstrates how antimicrobial treatments reduce microbial diversity and alter sediment biogeochemistry. A few recommendations:
Improve the clarity of English in certain sections (especially Discussion), where syntax and transitions between ideas can be refined.
Where possible, reduce redundancy in the Discussion, particularly in repetitive descriptions of diversity loss.
The inclusion of resistance gene profiling (e.g., ARGs) would have further strengthened the implications for public health, but this can be mentioned as a limitation or future direction.

The English could be improved to more clearly express the research.
Author Response
The manuscript presents a well-designed and rigorously executed metagenomic study evaluating the ecological impact of florfenicol in aquaculture sediments. It successfully demonstrates how antimicrobial treatments reduce microbial diversity and alter sediment biogeochemistry. A few recommendations:
Response: We sincerely thank the reviewer for the positive evaluation of our manuscript and for acknowledging the design and execution of our study. We are pleased that the ecological relevance and scientific rigor of our work were appreciated.
Regarding the reviewer’s recommendation to improve the language and clarity of the manuscript, we would like to confirm that we have thoroughly revised the entire text. To ensure the highest quality of written English, the manuscript was professionally edited by a native English speaker through an external editing service. We believe this has significantly enhanced the clarity and readability of our work.
- Improve the clarity of English in certain sections (especially Discussion), where syntax and transitions between ideas can be refined.
- Response: The manuscript has been thoroughly reviewed and edited by a professional scientific translation service. A certificate of translation is included to ensure the clarity and quality of the English throughout the text (see the attached PDF file), particularly in the Discussion section.
- Where possible, reduce redundancy in the Discussion, particularly in repetitive descriptions of diversity loss.
- Response: The change was made within the text.
- The inclusion of resistance gene profiling (e.g., ARGs) would have further strengthened the implications for public health, but this can be mentioned as a limitation or future direction.
- Response: Thank you for this valuable observation. We agree that including resistance gene profiling (e.g., ARGs) would have significantly strengthened the public health relevance of the study. As such, we have incorporated this point into the revised manuscript as a limitation and a potential direction for future research, highlighting the value of complementary metagenomic analyses.

Round 2
Reviewer 2 Report
Comments and Suggestions for Authors
The authors have diligently revised the manuscript. Additional information sought by me have been incorporated.
Author Response
We sincerely thank you for your positive comment and for approving the manuscript.